# The Position of *Lophozia dubia* (Lophoziaceae, Marchantiophyta) in the Phylogenetic System of *Lophozia* and the Distribution of *Lophozia* in Southeast Eurasia, Extending to Indonesia

**DOI:** 10.3390/plants13030367

**Published:** 2024-01-26

**Authors:** Vadim A. Bakalin, Yulia D. Maltseva, Aleksey V. Troitsky

**Affiliations:** 1Laboratory of Cryptogamic Biota, Botanical Garden-Institute FEB RAS, Makovskogo Street 142, Vladivostok 690024, Russia; maltseva.yu.dm@gmail.com; 2Belozersky Institute of Physico-Chemical Biology, Lomonosov Moscow State University, Leninskie Gory 1, Moscow 119991, Russia

**Keywords:** *Lophozia*, Lophoziaceae, molecular genetics, *trn*L–F, ITS 1–2, Southeast Asia, East Asia

## Abstract

*Lophozia pallida*, the commonly used name for a rare and little-known Sino-Himalayan species, was found to be a synonym of *Lophozia dubia*, a forgotten and previously misinterpreted species known in Indonesia. A comparative study of herbarium materials and our collections made it possible to ‘extend’ the distribution of *Lophozia* s. str. southward to Indonesia. The description of oil bodies from the species is provided for the first time. The position of the species in the *Lophozia* phylogenetic system demonstrates its clear differences from the morphologically similar *Lophozia guttulata* and its phylogenetic relationship with the Japanese–Korean *Lophozia koreana*.

## 1. Introduction

*Lophozia* s.l. is recognized as one of the most complex genera series in the northern Holarctic. Since the first description of this group [1,2], it has attracted a lot of attention. In the 20th century, a significant number of works were devoted to this group [3,4,5,6,7,8,9,10,11]. The widespread use of molecular genetic methods has breathed new life into our understanding of the morphology and taxonomy of the group. *Lophozia* (Dumort.) Dumort., after the transfer of alien elements to other genera and even families [12,13], turned out to be a relatively small genus, encompassing, according to the latest list, 16 species [14]. Meanwhile, (1) the species status of some of these species may be questioned, and (2) *Lophozia longiflora* (Nees) Schiffn is regarded, in the latter list, as a possible synonym of *L. guttulata* (Lindb.) A. Evans, which was shown to be an independent taxon by Vilnet et al. [15]. To this number must also be added the species that have been described more recently:

(1) *Lophozia fuscovirens* Bakalin & Vilnet, apparently an Arctic–subarctic species, so far known from Northern Okhotia (the Russian Far East) and the Spitsbergen Archipelago [16,17];

(2) *L. svalbardensis* Konstant., Vilnet & Mamontov, so far known only from Spitsbergen Archipelago [18];

(3) *L. obscura* (Bakalin) A. V. Troitsky, Bakalin & Fedosov, initially described as *Schistochilopsis* (N. Kitag.) Konstant., and then, after molecular genetic analysis, transferred to *Lophozia*, known from the Southern Kuril Islands [19];

(4) *L. koreana* (Bakalin, S.S. Choi & B.Y. Sun) Maltseva, Vilnet & Bakalin, originally described as *Tritomaria*, and then, based on molecular genetic research, transferred to *Lophozia*, known from Korea and Japan [20];

(5) *L. nepalensis* Bakalin, known from Nepal [21] and synonymized with *Lophoziopsis longidens* (Lindb.) Konstant. & Vilnet by Söderström et al. [14].

Thus, as of today, the genus includes from 20 to 22 species depending of writer estimations. The taxa of *Lophozia* occur mainly in the north of the Holarctic, where they play an important role in the formation of subarctic mossy synusia, especially in tundra communities. The range of the vast majority of these species does not pass 40 N in both the Western and Eastern Hemispheres. The only exception is a small group of species with a predominantly East Asian distribution, including: (1) *Lophozia lantratoviae*, which widely penetrates East Asia, at least as far as Chinese Sichuan, Japan and Korea [21,22,23]; (2) *Lophozia silvicola* H. Buch, identified southward of the Guizhou Province of China [24]. Whether this is the same species as that which is widespread in the north of the Holarctic is questionable; (3) *L. lacerata* N. Kitag., which reaches Taiwan [21,25]; (4) *Lophozia pallida* (Steph.) Grolle, known only from Yunnan Province of China, Bhutan and Nepal [14,26,27]; and (5) *Lophozia guttulata*, which reaches the Taiwan Province of China (Kitagawa [28] under *L. fauriana* Steph.).

According to available data, only one species, *Lophozia longiflora*, known from subantarctic (thus making it bipolar in distribution) has been found in South Georgia [25]. 

Among the taxa housed in the genus *Lophozia*, perhaps the most enigmatic is *Lophozia pallida*, for which several specimens are known from Sino-Himalaya. Moreover, the Yunnan Province of China houses two type localities of the taxa now known as the *L. pallida*: *Anastrophyllum pallidum* Steph. type (“Ma-eul-chan”, Delavay, G) and *Lophozia handelii* Herzog type (“Wahap” Handel-Mazzetti 1285, JE). The latter was previously interpreted by one of our group as a synonym of *Lophozia guttulata* [25], but Söderström et al. [14] believed that it belongs to a species distinct from *L. guttulata*, and is identical to *Anastrophyllum pallidum*. The cited authors synonymized both names, creating a new combination for the priority basionym (*Lophozia pallida*). Describing this situation, Söderström et al. ([14]: p. 29) wrote: “One of us (JV) has seen rich material of *Lophozia handelii* Herzog from Himalaya and it appears to be a well-developed form of *Lophozia guttulata*. However, it is in most cases possible to separate it from the latter and we therefore retain the species pending further studies”. Following this estimation, we reviewed the type materials and studied other herbarium specimens, and came to the same conclusion; namely, that *Lophozia pallida* is different from *L. guttulata*. New specimen of this species (*L. pallida*) was collected by us in the Yunnan Province of China in 2018. This gave us the opportunity to discuss its taxonomy again including, additionally, its molecular data.

The discussion of the systematic position and distribution of this taxon is the main goal of the present paper.

## 2. Results

### 2.1. Molecular Genetics

Two accessions (one of ITS1–2 and one of *trn*L–F) were produced and deposited into GenBank for the specimen *Lophozia dubia* Schiffn. C-83-7-18 (VBGI).

On both the ITS and *trn*L–F phylogenetic trees (Figure 1 and Figure 2) the *Lophozia dubia* specimen forms a sister branch to *Lophozia koreana.*

The distance between the *Lophozia dubia* specimen and other species of *Lophozia* is consistent with the level of interspecific variability in the genus. The divergence among taxa varies from 4.5 to 7.6% for ITS1–2 and from 0.3 to 5.9% for *trn*L–F (Table 1).

*Lophozia guttulata*, mentioned before as being morphologically similar if not identical to *Lophozia dubia* cf. [14,25], on the phylogenetic trees assumes a remote position from it, and the genetic distances between these two species are 6.4% for ITS and 3.4% for *trn*L–F. Thus, it can be concluded that these two species are genetically different from each other.

The *trn*L–F phylogeny is shown in Figure 1. The alignment of 51 *trn*L–F sequences consisted of 530 positions, among which 73 were parsimony-informative, 39 were singletons and 418 were constant sites. The base frequencies across all sites were A: 0.347, C: 0.154, G: 0.178 and T: 0.321. The ML criterion resulted in a consensus tree with a log likelihood of −1913.531. The arithmetic means of the log likelihoods from the Bayesian analysis of each sampling run were −1957.52 and −1957.17.

The *Lophozia* species forms five highly supported clades; however, the relationship between them cannot be reliably established. *Lophoziopsis* species are distributed into three clades.

The ITS1–2 phylogenetic tree shown in Figure 2 was constructed from an alignment of 50 sequences that contained 866 positions, of which 229 were parsimony-informative, 86 were singletons and 551 were constant sites. The base frequencies across all sites were A: 0.176, C: 0.290, G: 0.332 and T: 0.203. The ML criterion resulted in a consensus tree with a log likelihood of −4677.341. The arithmetic means of the log likelihoods from the Bayesian analysis of each sampling run were −4717.82 and −4717.17.

On the ITS tree, the *Lophozia* and *Lophoziopsis* species combine again into five and three highly supported clades, respectively, with no reliably established relationship between species of the same genus. In this case, the genus *Lophoziopsis* turns out to be paraphyletic and the position of *Lophozia longiflora* is changed.

The topologies of the two trees are similar, but not identical, so a combined ITS–trnL–F tree was constructed (Figure 3). The characteristics of the combined dataset are as follows: the alignment consists of 50 sequences with 1392 positions, of which 286 were parsimony-informative, 138 were singletons and 968 were constant sites. The base frequencies across all sites were A: 0.250, C: 0.250, G: 0.250 and T: 0.250. The ML criterion resulted in a consensus tree with a log likelihood of −6737.043. The arithmetic means of the log likelihoods from the Bayesian analysis of each sampling run were −6769.93 and −6767.65.

On the combined tree as well as on individual ones, *Lophozia dubia*, with high support, occupies a sister position to *Lophozia koreana*. The *p*-distances between the sequences of these two species here exceed the *L. koreana* infraspecific differences (Table 1). The genus *Lophosiopsis* is monophyletic, with PP = 0.99.

### 2.2. Morphology

Since the molecular genetic study involves material from one specimen only (the others are too old), it helped demonstrate that the taxon called *Lophozia pallida* is not the same as *L. guttulata*. Other results are based on comparative morphology. As it turned out, as a result of the herbarium studies, the following specimens belong to the same taxon, despite the fact that they are mentioned in the herbarium sheets under three different names:

China. “Yunnan bor.-occid.: in montis Wahap rope pagum Yungning regione frigide temperate infra casulam Maoniubi. Substr. trunc. Abietum; alt 3800–4030 m. 21.VII.1915 Handel-Mazzetti (Diar. Nr. 1285)” (Lectotype of *Lophozia handelii*, JE Nr. 7131 JE04009230, isolectotype JE04009229; isolectotype NICH 225494; isolectotype PC0102533). There are two specimens in JE that are presumably parts of the same original specimen, barcoded with two different numbers: JE04009229 and JE04009230. One of them, marked using a typing machine as Typus and recognized as a holotype in subsequent note (without an evidence base, because typus does not mean holotype by itself), while the other one that is larger in size has real handwriting by Herzog on its label, with the mark “n. sp.” We think the latter one (JE04009230) should be recognized as the lectotype and the other one JE04009229 should be treated as an isolectotype.

China. Yunnan Province. “Ma-cul-chan” Delavay (Lectotype of *Anastrophyllum pallidum* according to Söderström et al. [14] with reference to Bonner 1962: 77 [not seen] G-53383/10810).

China. Yunnan Province, Zhongdian, alt. 3520 m. 14.VI.1981 Li X.J. 920 (JE, as *Lophozia handelii*).

Indonesia. Sumatra Island. “Ad decliv. occid. montis ignivomi Merapi Infra craterem ad terram. Regio alpina, alt. +/− 2600 m. s. m. 31.VII.1894 (No. 605). (Lectotype of *Lophozia dubia*, selected here FH (Harvard University Herbaria 01142702 https://kiki.huh.harvard.edu/databases/specimen_search.php?barcode=01142702 (accessed on 28 December 2023), isolectotype G00121692.)

Nepal. Khumbu, Lobuche, 5100 m alt. Poelt H77, Poelt H91 (JE, as *Lophozia handelii*).

Nepal. Langtang, Brambing, 4400 m alt. Scree N-exp. (*Salix* shrub): ground mosses. Coll G. & S. Miehe 09 10 1986 (JE, as *Lophozia handelii*).

Nepal. South of Keldang, Dupku Danda, alt 4120 m. Hab. north-facing *Rhododendron* L. thicket. Coll G. & S. Miehe 6993d 28.VII.1986 (JE, as *Lophozia handelii*).

China. Yunnan Province. Shangri-La County, Xiao-Zhong-Dian Xiang, Tian-Bao Mountain. Narrow valley with coniferous forest with admixture of rhododendron and many limestone outcrops (also resulted in basic reaction of humificated soil). Partly shaded moist lying trunk of living rhododendron. 16.10.2018 V.A. Bakalin & W.Z. Ma C-83-7-18 VBGI-88807.

In addition to the mentioned specimens, Grolle [26] provided the following reports for *Lophozia handelii*: Nepal “Mahalangur Rimal, Khumbu, ostlich Khumzung gegen das Dudh Kosi-Tal, 3900–4000 m, leg. Poelt, 1962, Nr. H 54. Vorhimalaya, *Abies* Mill.-*Rhododendron*·Wald um Thodung, 3000 m, leg. Poelt, 1962, Nr. H 214.” (l.c.: 277).

Additionally, Long and Grolle [27] provide the subsequent data for the distribution of the *Lophozia handelii* in Nepal C-THIMPHU: valley above Chenkaphug, 3300 m, L8804; below Chelai La, Paro Chu, 3150m, Ll0974; PUNAKHA: Pele La summit, 3450 m, L7920; MONGAR: Thrumse La, 3600 & 3680 m, L8458b, 8699. They also provide the habitat for *Lophozia handelii*: “on tree trunks and fallen logs in moist mixed coniferous (*Abies/Juniperus* L./*Picea* A. Dietr./*Tsuga* (Endl.) Carrière/*Rhododendron*) forests”.

As molecular genetic research has shown, plants originally identified as *Lophozia pallida* are clearly genetically different from *Lophozia guttulata* and the ‘independence’ of this species can be considered confirmed. The remaining conclusions are based on comparisons of plant morphology. The main result is that not only are the specimens initially assigned to *Lophozia pallida* and *L. handelii* morphologically similar, but so is the identity of the type materials of *Lophozia dubia*. This, firstly, changes our ideas about the general range of the taxon, which, as it turns out, extends to Java and Sumatra, but it also raises the question of the priority name for the species, since the oldest epithet is *Lophozia dubia*.

## 3. Discussion

*Lophozia dubia* was described by Schiffner [29] from one specimen from the island of Java and two specimens from the island of Sumatra. However, this name does not appear in recent checklists. For instance, in the checklist of the liverworts of Java [30], this name appears as a synonym of *Hattoriella subcrispa* (Herzog) Bakalin (=*Mesoptychia subcrispa* (Herzog) L. Söderstr. & Váňa in later interpretation following to Váňa et al. [31]). In the international Catalogue of Life database, *Lophozia dubia* (https://www.catalogueoflife.org/data/taxon/425N9, accessed on 20 November 2023) also appears as a synonym of *Mesoptychia subcrispa*, with reference to the same checklist of Java liverworts. In this case, however, the question arises as to why *Lophozia dubia* appears to be synonymous with *Lophozia subcrispa* Herzog, but not vice versa, since *L. subcrispa* was first described 45 years later [32]. Moreover, no formal synonymization was made in the checklist of Java liverworts, and the origin of mentioned synonymy treatment is unclear to us. The difference between the plants in the type specimen *Lophozia subcrispa* (typical *Mesoptychia–Hattoriella*) and the type specimen *Lophozia dubia* (typical *Lophozia* s. str.) is very striking; we studied the holotype of *Lophozia subcrispa* JE040006259 and isotypes in PC0103172 and JE04006260, images of the species are provided in Bakalin [33]. The prominent worldwide expert on liverworts V. Schiffner [29] describes the relationships of *Lophozia dubia* as follows (translated by us from Latin, l.c.: 203): “This tropical plant is very closely related to our native *Lophozia ventricosa* (Dicks.) Dumort., and perhaps scarcely different from it; more serious problems are desired by the sexual organs as yet unknown. Be that as it may, in the meantime it should be regarded as a proper species, for the sin is more in confounding than in discriminating the knowledge of the nature of things.” (Original: Arctissime affinis *Lophoziae ventricosae* nostrati et forsitan ab illa vix diversa est haec planta tropica; discrimina gravioris momenti desiderantur ab organis sexualibus adhuc ignotis. Utut sit, interea pro specie propria habeatur, peccatur enim magis in confundendis quam in discriminandis rerum naturae cognitionibus.) It seems extremely doubtful that Schiffner could mistake *Mesoptychia* (in the modern sense) for *Lophozia ventricosa* (in the broad interpretation accepted at the time of his writing the work), but not for those species known to him, for example, *Lophozia collaris* (Nees) Dumort. or *L. bantriensis* (Hook.) Steph., to which *Lophozia subcrispa* is likely morphologically similar.

Based on our observations and the lectotypification carried out here, the name *Lophozia dubia* takes priority over both *Anastrophyllum pallidum* and *Lophozia handelii*, and the following synonymization is necessary:

*Lophozia dubia* Schiffn. Denkschriften der Kaiserlichen Akademie der Wissenschaften, Wien. Mathematisch-naturwissenschaftliche Klasse 67: 202. 1898. (Denkschr. Kaiserl. Akad. Wiss., Wien Math.-Naturwiss. Kl.) Java: Prov. Preanger. In regione superiore montis ignivomi Gedeh ad terram infra craterem. Regio alpina, alt. ± 2700 m s. m.—10.7.1894 (in caespitibus Aploziae Stephani). (No. 603.). Sumatra occid.: Ad decliv. occid. montis ignivomi Merapi in silva primaeva ad terram. Regio nubium, alt. 1760 m s. m.—30.7.1894 (…). (No. 604); ibidem: infra craterem ad terram. Regio aplina, alt ± 2600 m s. m.—31.7.1894. (No. 605) leg. Schiffner. The last specimen (No. 605) is selected here as the lectotype.

=*Anastrophyllum pallidum* Steph., Bull. Herb. Boissier ser. 2, 1: 1131 (Sp. Hepat. 2: 114), 1901 (*Lophozia pallida* (Steph.) Grolle, J. Jap. Bot. 39: 174, 1964) syn. nov.=*Lophozia handelii* Herzog, Symb. Sin. 5: 14, 1930 syn. nov.

This species has been identified across 14 specimens (most of them studied by our group), but its ecology is still not well known. Apparently, the species grows both on soil and on decaying wood and even on *Rhododendron* trunks in the upper part of the forest belt (presumably including *Rhododendron* forests) and above it. This species is known from epixylous and epigean habitats in the Sino-Himalayas, but in Indonesia it is still found only on soil. On soil, the species’ companions are representatives of the genus *Scapania*, *Solenostoma stephanii* (Schiffn.) Steph. (Indonesian materials), while on decaying wood in the Yunnan Province of China, this species is accompanied by taxa of a broader ecology, growing both on crystalline substrates and on decaying wood: *Mylia vietnamica* Bakalin & Vilnet and *Anastrepta orcadensis* (Hook.) Schiffn.

The distance from Nepal to West Java is more than 4500 km. The species, across this range, probably occurs very sporadically (possibly with the exception of the eastern Himalayas), in high elevations in the mountains. Existing populations are separated by significant water barriers and lower-elevation tropical communities. The genetic uniformity of the species may be questioned, since individual populations are likely to exist in complete isolation for a long time. It seems appropriate to discuss the variability of the species based on a comparison of the descriptions made from different specimens. For this purpose, below we provide their morphological descriptions, mostly compiled by us, but first we will start with the original description by Schiffner [29].

Schiffner ([29]: p. 174) provides a fairly detailed description in Latin, which we present here in our English translation: Sterile. Within tufts of terricolous liverworts (*Aplozia stephanii* Schiffn., *Scapania*, etc.), pale green in appearance again dichotomous, in fact with lateral branches arising from the axils of the leaves. The creeping stem is furnished with long pale rootlets from the bottom to the tip, the tips of the branches are pale brown, densely leafy up to 2 cm long with flattened leaves up to 2.8 mm wide. Leaves at base broad obliquely inserted subexplanate or canaliculate broad-ovate 1.6 mm long, 1.3 mm wide, bilobed (rarely trilobed), sinus +/− 1/4 of length obtuse, subequal lobes (ventral slightly larger) broadly trigonous acute. Cells elliptic, barely sinuate with visible triangles, 0.03 × 0.023 mm. The terminal leaves produce pale unicellular buds at the tips of the lobes. No amphigastria. Otherwise unknown. (Latin original: Sterilis. Inter caespites Hepaticarum terricolarum (*Aploziae stephanii*, *Scapaniae* etc.) repens, pallide viridis ad speciem iterum dichotoma re vera ramis lateralibus ex axilla folii ortis. Caulis repens subtus ad apicem usque radicellis longis pallidis instructus, fuscus apices ramorum versus pallidus, dense foliatus ad 2 cm longus cum foliis explanatis ad 2.8 mm latus. Folia basi lata oblique inserta subexplanata vel canaliculata late-ovata 1.6 mm longa, 1.3 mm lata, biloba (rarissime triloba), sinus +/− 1/4 longitudinis obtuso, lobis subinaequalibus (ventrali paulo majore) late trigonis acutiusculis. Cellulae ellipticae vix sinuatae trigonis conspicuis 0.03 × 0.023 mm. Folia terminalia ad apices loborum gemmas unicellulares pallidas proferunt. Amphigastria nulla. Cetera non vidi.) (Figure 4).

Our description, based on the fresh material from the Yunnan Province of China (C-83-7-23, as cited above): Plants prostrate to ascending near the apices, bright green when fresh and greenish to yellowish greenish in herbarium, somewhat rigid, mixed with other liverworts, 1.8–4.0 mm wide and 10–25 mm long. Rhizoids abundant, mostly in unclear, obliquely spreading and intertwined fascicles, grayish brownish. Stem slightly flexuous to almost straight, dorsal and lateral sides yellowish greenish, ventral side vinaceous brown to blackish brown; stem cross-section nearly rounded, 300–450 µm in diameter or slightly transversely ellipsoidal, outer cells on lateral and dorsal sides 25–30 µm in diameter, nearly thin-walled to walls slightly thickened, external wall always distinctly thickened, trigones moderate in size, mostly concave, outer cells on ventral side slightly smaller and more thin-walled; inner cells thin-walled, 25–30 µm in diameter, with small trigones, microcellous layer occupying (1/2–)2/3 of stem thickness, composed of cells 10–20 µm in diameter, very thin-walled and with vestigial to virtually absent trigones. Leaves subtransversely inserted, sheathing the stem in lower third of the leaf length (leaf base semi-funnelate), above concave-canaliculate, obliquely to erect spreading in lateral direction or slightly to strongly turned dorsally; contiguous; when flattened in the slide obliquely ovate to widely obliquely ovate, 1000–2250 × 1000–2380 µm, divided by crescentic to widely V-shaped sinus descending to 1/10–1/4 of leaf length into 2 unequal lobes, lobes triangular to widely so, acute to obtuse. Midleaf cells thin-walled, subisodiametric to shortly oblong, 25–38 × 20–33 µm; trigones moderate in size, convex; oil bodies in the midleaf cells 9–16 per cell, granulate, spherical, 4–8 µm in diameter to ellipsoidal, sometimes irregularly so, 3–5(–8) × 4–10(–12) µm; basal cells oblong, 30–45 µm long; cells along leaf margin smaller than in next rows inward, elongate along margin or isodiametric, 20–30 µm along margin, nearly thin-walled with convex, moderate in size to large trigones, outer wall noticeably thickened, cuticle smooth throughout. Dioicous. Androecia not seen. Female bracts similar in size to large leaves. Bracteole absent. Perianth obovate, 1.5 mm long, plicate ion upper third, gradually narrowed to the not-beaked mouth, mouth dentate-ciliate with cilia up to 3 cells long. Sporophyte not seen (Figure 5 and Figure 6).

Description of plants in lectotype of *Lophozia handelii* (herbarium number Nr. 7131, JE04009230): Plants pallid, more or less soft, 1.5–2.0 mm wide, sparsely branched; stem pale brownish, ventral side brown, microcellous layer 1/2 or less of the stem height; rhizoids numerous, obliquely spreading or forming a mat under the stem; leaves subtransversely inserted, shallowly sheathing the stem very near to the base and obliquely spreading and oriented above, nearly canaliculate, with some leaf lobes deflexed, when flattened in the slide then subrectangular to rectangular–ovate, widest near the middle, 1.0–1.3 × 0.8–1.1 mm, divided by U-shaped sinus descending to 1/4–1/3 of leaf length into two somewhat obtuse subequal or only slightly unequal lobes (Figure 7 and Figure 8).

Description based on holotype of *Anastrophyllum pallidum* G00053383/10810 (note, the holotype contains only one plant without the apical part even): Plant 2.2 mm wide, of unknown length, yellowish brownish in color. Stem brownish with purple–violet–brown-colored ventral side. Rhizoids numerous, colorless to brownish, erect to obliquely spreading in unclear fascicles or forming a mat under the stem. Leaves obliquely inserted, strongly second dorsally and turned to dorsal side of the shoot (giving an ‘anastrophylloid’ appearance), canaliculate-concave, ca. 1.6–2.0 × 1.6–1.9 mm, divided by U-shaped sinus descending to 1/5 of leaf length into two unequal acute lobes, ended by 1–2 superposed cells, that gives the look of *Tritomaria* (noted by Kitagawa). Midleaf cells subisodiametric 35–45 × 35–40 µm, thin-walled, trigones large, convex (Figure 9A–G).

Description based on isolectotype of *Lophozia handelii* in PC102533: Plants ascending, yellowish brownish, merely rigid, 1.65–2.125 mm wide and 10–15 mm long (length measurements may be incorrect because of limited material). Rhizoids numerous, nearly colorless, obliquely to erect spreading in unclear fascicles or forming a mat under the stem. Leaves subtransversely to almost transversely inserted, obliquely spreading, subtransversely oriented, concave-canaliculate, ‘canal line’ slightly arcuate, obliquely elliptic, secund ventrally (ventral lobe larger), 950–1000 × 800–1050 µm, divided by U- to V-shaped sinus descending to 1/4 of leaf length into two subequal to unequal triangular lobes with axis subparallel to divergent. Midleaf cells 20.0–32.5 × 20.0–27.5 µm, thin-walled, trigones moderate in size, slightly concave or slightly convex, cuticle smooth. Gemmae colorless, very few in number, produced in the apices of the uppermost leaves, 17–19 × 13–17 µm, 2-celled, 4–6-angular in the projection (Figure 9H–R).

Description based on isolectotype of *Lophozia dubia* G00121692. Plants ascending 1.0–1.5 µm wide and 7–15 mm long, yellowish brownish, growing as tiny admixture to other hepatics. Stem yellowish brownish, ventral side purple–brown, slightly transversely elliptic in the cross section, 200–260 × 260–320 µm, microcellous layer 1/3–2/5 of stem height, main tissue cells irregular in shape, 28–40 µm in diameter, with moderate to small concave trigones, thin-walled, external cell walls of the outer layer of the cross-section merely thickened, microcellous layer cells thin-walled, with vestigial trigones, 16–25 µm in diameter. Rhizoids numerous, colorless to brownish, with violet–purple tint at very beginning near the stem. Leaves contiguous to subimbricate and then enclosed to one another, obliquely inserted, obliquely spreading, obliquely to subtransversely oriented, concave-canaliculate, not or very loosely sheathing in the base, 0.7–0.9 × 0.6–0.7 mm, nearly rectangular to ovate and obovate, divided by V- to U-shaped sinus, descending to 1/4 of leaf length into two unequal to subequal triangular, acute lobes. Midleaf cells thin-walled, subisodiametric to shortly oblong, 28–40 × 26–36 µm, trigones moderate in size, convex, cuticle loosely (but not finely) papillose (Figure 10A,B,F).

Additionally, here we provide photographs of the specimens taken by Miehe 6993d (Figure 10C–E,G,H and Figure 11A–C) and Li X.J.920 (Figure 11D).

As can be seen, the overall size of the plants varies quite significantly from specimen to specimen. The authentic materials of *Lophozia handelii* and *L. dubia* contain smaller plants than the *Anastrophyllum pallidum* holotype. The specimen collected in China (C-83-7-18) is closer to the upper limit of plant sizes seen (thus close to those of the *Anastrophyllum pallidum* type), although the variability in the size of plants in this specimen overlaps greatly with other materials. Other materials studied, cited in the results section, occupy mostly an intermediate position. At the same time, the nature of the leaf cell (distinctly trigonous), and the size of the cells, remains more or less stable from specimen to specimen, overlapping to a large extent. One of the features subject to great variation is the leaf shape, which is sometimes very similar to *L. guttulata*—relatively narrow and merely deeply divided, whereas sometimes the leaves are as wide as they are long and only shortly divided by the sinus. This partly explains the earlier approach to the synonymy of two names [25]. However, this variation sometimes occurs within one plant alone and, as a rule, is observable within one more or less large populations. We hypothesize that the observed variation is environmentally induced.

The general differentiating characteristics of *Lophozia dubia* from other *Lophozia* taxa are a semi-funnellate (nearly sheathing) leaf base, covering the stem in the lower third of the leaf length, dorsally turned leaves with unequal lobes (the ventral is larger), and, usually (except for in small plants), leaves canaliculate-concave in the upper half. A particular challenge is distinguishing the species from *Lophozia lacerata*, whose range extends from the northern parts of the Russian Far East to the Taiwan Province of China. This species has common features with *L. dubia*, including dorsally secund leaves with unequal lobes (the ventral is larger), and an absence of female bracteole (where a bracteole absence is a rare feature in *Lophozia* in general). However, *L. lacerata* is characterized by smaller cells in the midleaf, varying from 18 to 25 µm wide, but usually not exceeding 22 µm (in this sense, the cell size in *L. lacerata* is somewhat similar to *Pseudolophozia sudetica* (Nees ex Huebener) Konstant. & Vilnet), while, in *L. dubia*, cells usually wider than 25 µm in the midleaf. The semi-funnellate leaf base, which is always present in well-developed *L. dubia* plants, is also not characteristic for *L. lacerata*. Another feature that distinguishes both species is the structure of the perianth mouth armature, which is torn and long-laciniate in *L. lacerata*, while dentate-ciliate in *L. dubia*, with cilia up to 3 cells long. *Lophozia dubia* shows close genetic relationships to *L. koreana*, although these species are distinctly different. *Lophozia koreana* was originally described as *Tritomaria* due to its superficial resemblance to *T. mexicana* Bakalin and, indeed, it is more similar to *Tritomaria* than to *Lophozia* due to its strongly unequal leaf lobes. The lobes’ inequality is somewhat inherent to *L. dubia* too, although to a much lesser extent than is common in *Tritomaria* and *Lophozia koreana*. Other differentiation features of *L. koreana* include a striolate-papillose leaf cuticle (smooth in *L. dubia*) and pink unicellular gemmae (versus 1–2-celled and greenish to colorless in *L. dubia*). The proper illustrations were published by Bakalin et al. [34]. The genetic distances between two taxa justify the difference of the species rank between the two taxa as well (Table 1).

## 4. Materials and Methods

### 4.1. Morphological Studies

The traditional morphological method was used to study specimens of some representatives of *Lophozia* from East and Southeast Asia in the herbaria JE, G, NICH, FH, PC and VBGI (international acronyms following the Index Herbariorum [35]). In the course of working in herbaria, as a rule, morphological descriptions are prepared, accompanied by photographs. Photographs were taken with a handheld portable digital camera through a microscope ocular, or using special cameras mounted on microscopes. Subsequently, using photographs covered with translucent paper or using a drawing apparatus mounted on an Olympus SZX10 microscope (at VBGI), drawings of plants were made from a number of specimens. All photographs and drawings have been placed in their corresponding places in the text. In our study of living plants collected from Yunnan (VBGI-88807, detailed below), we photographed the oil bodies in leaf cells, and therefore obtained the first data on the oil bodies in the species.

### 4.2. Molecular Genetic Research

#### 4.2.1. Taxon Sampling

To compile the dataset for molecular phylogenetic analysis, we studied one specimen of *Lophozia pallida* (*L. dubia*), specimen voucher C-83-7-18 (VBGI), from the Chinese Yunnan Province. Our alignment was based on the dataset from our previous study [20], except for species of the family Scapaniaceae. For tree rooting we selected *Obtusifolium obtusum* (Lindb.) S.W. Arnell from Obtusifoliaceae Bakalin & Fedosov following the recent phylogeny for the group in [36]. Specimen voucher details, including GenBank accession numbers, are listed in Table 2.

#### 4.2.2. DNA Isolation, Amplification and Sequencing

Two markers, the nuclear ITS1–2 and plastid *trn*L–F region, were used for a phylogenetic study. DNA was extracted from dried liverwort tissues using the DNeasy Plant Mini Kit (Qiagen, Hilden, Germany). Amplification of ITS1–2 and *trn*L–F was performed using an Encyclo Plus PCR kit (Evrogen, Moscow, Russia) with the primers listed in Table 3.

The polymerase chain reaction was performed in a total volume of 20 µL, including 1 µL of template DNA, 0.4 µL of Encyclo polymerase, 5 µL of Encyclo buffer, 0.4 µL of dNTP-mixture (included in Encyclo Plus PCR Kit), 13.4 µL (for *trn*L–F)/12.4 µL (for ITS1–2) of double-distilled water (Evrogen, Moscow, Russia), 1 µL of dimethylsulfoxide/DMSO (for ITS1–2) and 0.4 µL of each primer (forward and reverse, at a concentration of 5 pmol/µL). Polymerase chain reactions were carried out using the protocols for amplification listed in Table 4.

Amplified fragments were visualized on 1% agarose TAE gels via EthBr staining and purified using the Cleanup Mini Kit (Evrogen, Moscow, Russia). The DNA was sequenced using the ABI PRISM^®^ BigDye™ Terminator Cycle Sequencing Ready Reaction Kit (Applied Bio-systems, Waltham, MA, USA) with further analysis of the reaction products following the standard protocol on an automatic sequencer 3730 DNA Analyzer (Applied Bio-systems, Waltham, MA, USA) in the Genome Center, Engelhardt Institute of Molecular Biology, Russian Academy of Sciences, Moscow.

#### 4.2.3. Phylogenetic Analyses

Two datasets were compiled for the ITS1–2 and *trn*L–F loci and aligned using MAFFT [39,40,41] with standard settings, and then edited manually in BioEdit ver. 7.2.5 [42]. All positions of the final alignment were included in the phylogenetic analyses. Absent data at the ends of regions and gaps were treated as missing data.

Phylogenetic trees were reconstructed using two approaches: maximum likelihood (ML) [43] with IQ-tree ver. 1.6.12 [44] and Bayesian inference (BA) [45] with MrBayes ver. 3.2.7 [46].

For the ML analysis, the best-fitting evolutionary model of nucleotide substitutions according to the BIC value was HKY+F+G4 for the *trn*L–F dataset, TN+F+G4 for the ITS1–2 and TNe+G4 for the combined ITS–*trn*L–F dataset, determined via ModelFinder (model selection method which is implemented in IQ-tree) [47]. Consensus trees were constructed with 1000 bootstrap replicates.

Bayesian analyses were performed by running two parallel analyses using the GTR+I+G model. The analysis consisted of four Markov chains. Chains were run for five million generations, and trees were sampled every 500th generation. The first 2500 trees in each run were discarded as burn-in; thereafter, 15,000 trees were sampled from both runs to produce the resulting tree. Bayesian posterior probabilities were calculated from the trees sampled after burn-in. The average standard deviation of split frequencies between two runs reached 0.004 in *trn*L–F, 0.002 in ITS1–2 and 0.001 in ITS–*trn*L–F before the analysis was stopped.

The sequence variability among the specimens of the genus *Lophozia* was evaluated via the *p*-distances for each DNA locus in Mega 11 [48] using the pairwise deletion option for counting gaps.

## 5. Conclusions

In total, there are 20 to 22 species within the genus *Lophozia*, as discussed in the introduction. The many of them belong to a group of poorly morphologically differentiated taxa (*Lophozia wenzelii* (Nees) Steph., *L. longiflora*, *L. guttulata*, *L. ventricosa*, *L. murmanica* Kaal., *L. silvicola*) widely distributed in the northern Holarctic, whose status requires serious study, including molecular genetic research based on a larger number of specimens and a larger number of DNA loci. On the other hand, species occupying a geographically marginal position in the *Lophozia* generic area, such as *Lophozia fuscovirens*, *L. svalbardensis*, *L. koreana*, *L. dubia* and *L. lantratoviae*, are well differentiated both morphologically and genetically. It is necessary to add to this list *L. nepalensis* and *L. lacerata*, which have not been studied using genetic methods, but which are also morphologically distinguishable from the remaining bulk of the *Lophozia* taxa. In the list of these species, one can discern a serious influence from East Asia (the main range of all listed, except for the first two). This also allows us to conclude that the study of the diversity of *Lophozia* in the mountainous regions of East Asia should be continued, since it may lead to the discovery of species new to science that have special characteristics hitherto unknown about within the genus. The position of *L. nepalensis* is of special interest. As was mentioned above, it was synonymized with *Lophoziopsis longidens*, which is difficult to maintain. The distribution of gemmae and the leaf insertion and orientation in *Lophozia nepalensis* are very different from *Lophoziopsis* Konstant. & Vilnet in general. However, at present it is hardly possible to add anything to the characteristics mentioned in the original description, since the species is still known from this type only, and the status of *Lophozia nepalensis* is still questionable, partially due to a possible relationship with *Lophozia lantratoviae* Bakalin. This is one of the possible issues for future studies. Aside from the mentioned group of Sino-Himalayan *Lophozia,* the distribution of *Lophozia* in the subantarctic may be of special interest. The identity of the plants from South Georgia labelled as *L. longiflora* by Bakalin [25] remains to be verified; they may be a distinct species, only morphologically similar to the mostly hemiarctic *L. longiflora*.

## Figures and Tables

**Figure 1 plants-13-00367-f001:**
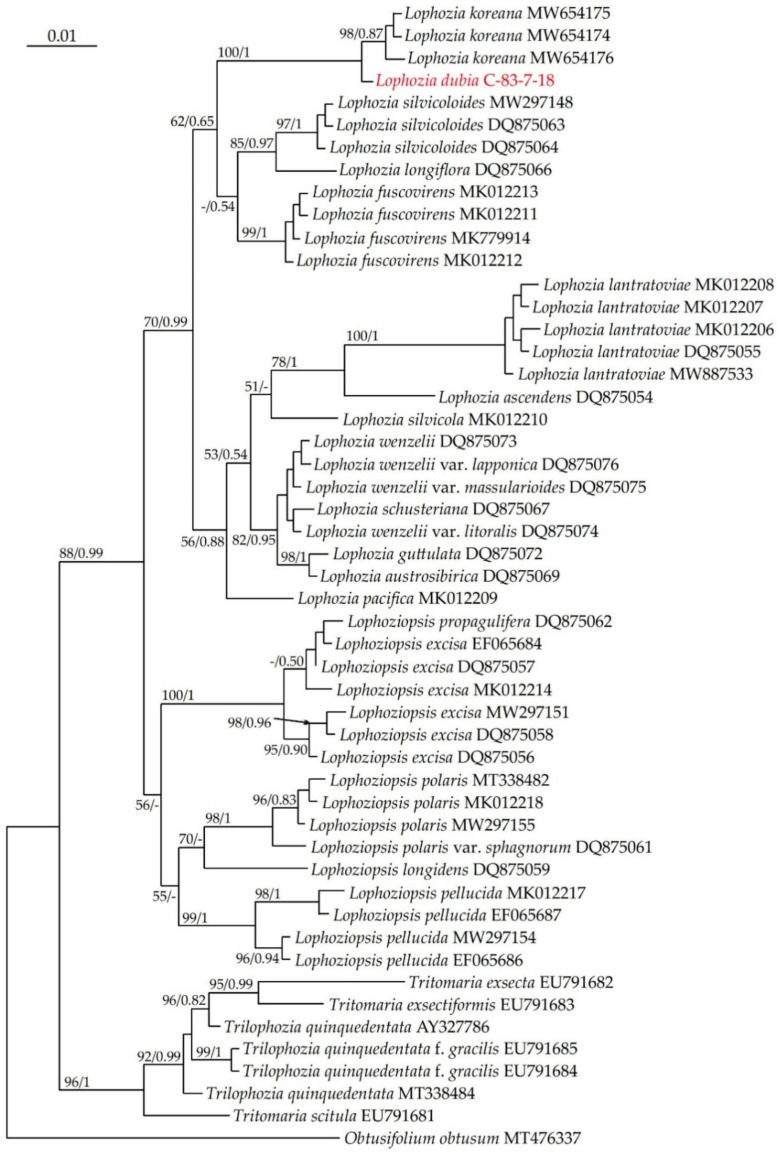
Phylogram obtained from a Bayesian analysis of the genus *Lophozia* (Dumort.) Dumort., *Lophoziopsis* Konstant. & Vilnet and related taxa based on *trn*L–F. Newly obtained specimen is marked in red. Bootstrap support values > 50% in ML analysis and Bayesian posterior probabilities PP > 0.50 are indicated. Scale bar denotes the number of nucleotide substitutions per site.

**Figure 2 plants-13-00367-f002:**
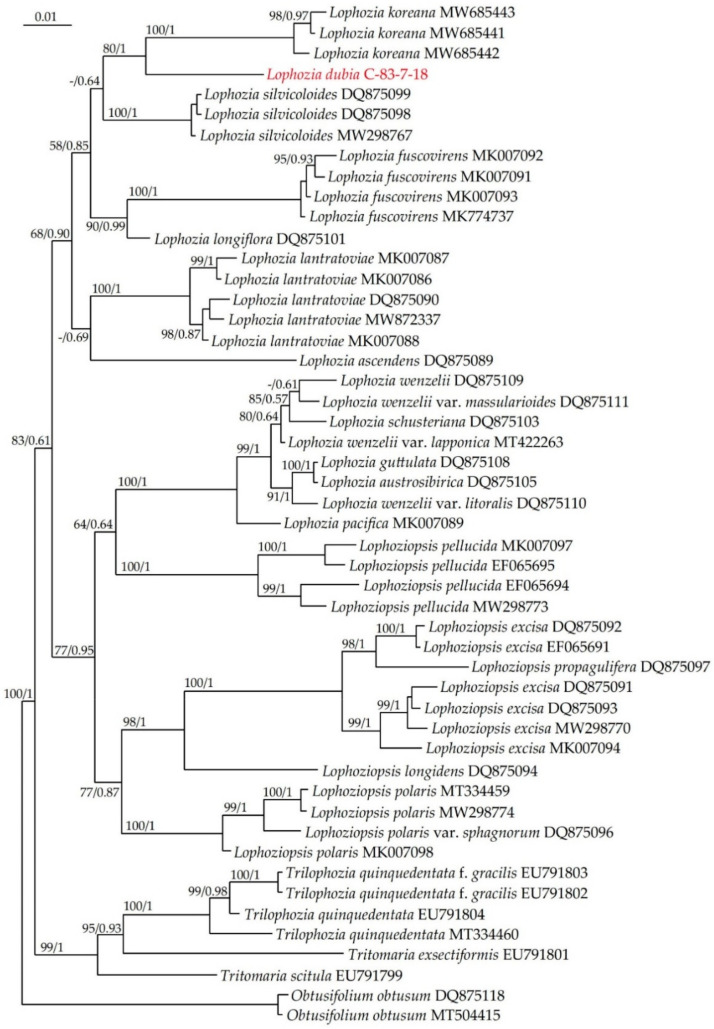
Phylogram obtained from a Bayesian analysis of the genus *Lophozia* (Dumort.) Dumort., *Lophoziopsis* Konstant. & Vilnet and related taxa based on ITS 1–2. Newly obtained specimen is marked in red. Bootstrap support values > 50% in ML analysis and Bayesian posterior probabilities PP > 0.50 are indicated. Scale bar denotes the number of nucleotide substitutions per site.

**Figure 3 plants-13-00367-f003:**
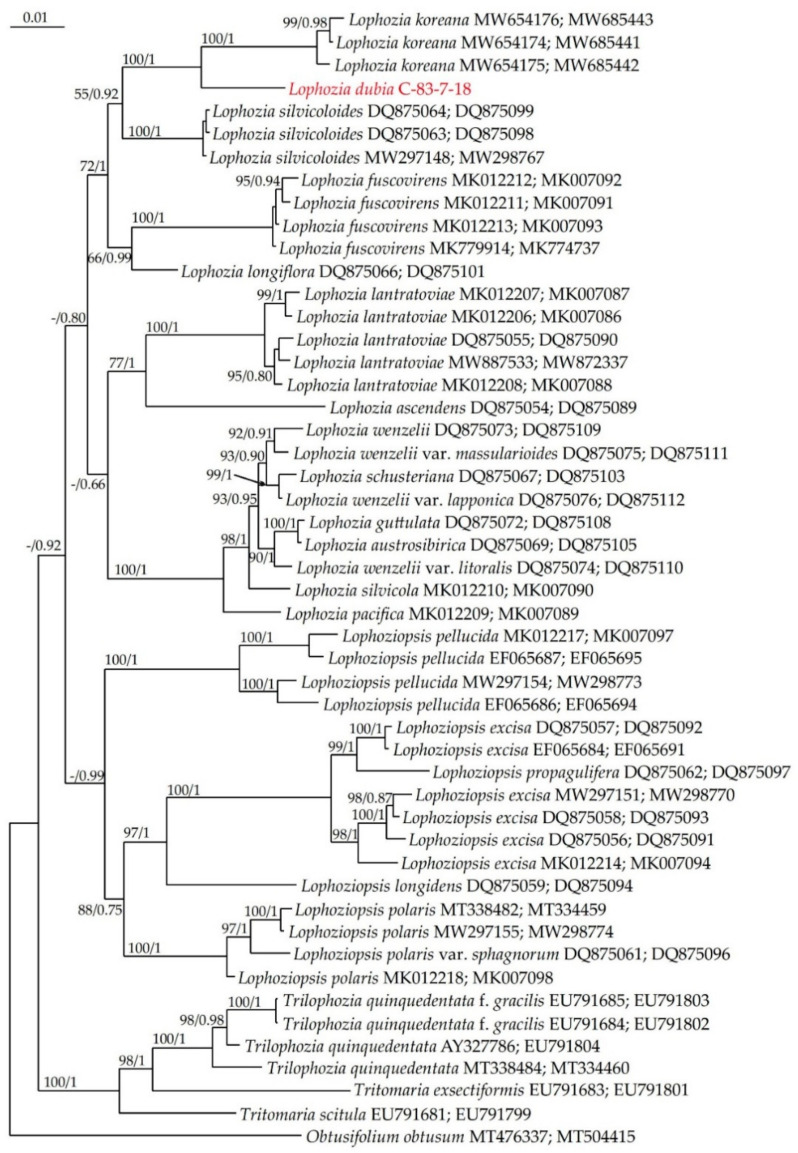
Phylogram obtained from a Bayesian analysis of the genus *Lophozia* (Dumort.) Dumort., *Lophoziopsis* Konstant. & Vilnet and related taxa based on a combined ITS–*trn*L–F dataset. Newly obtained specimen is marked in red. Bootstrap support values > 50% in ML analysis and Bayesian posterior probabilities PP > 0.50 are indicated. Scale bar denotes the number of nucleotide substitutions per site.

**Figure 4 plants-13-00367-f004:**
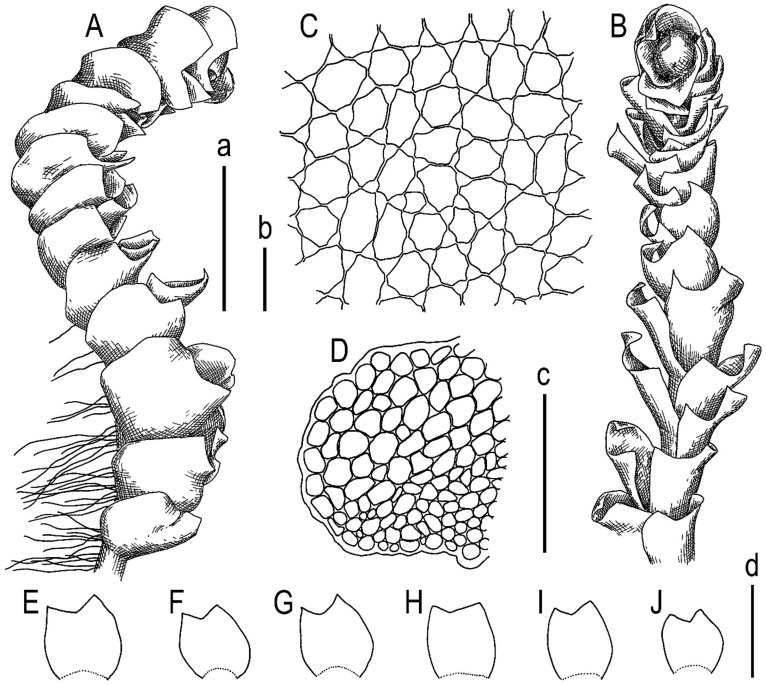
*Lophozia dubia* Schiffn. (**A**) upper part of a shoot fragment, side view; (**B**) upper part of a shoot fragment, dorsal view; (**C**) middle cells of leaf; (**D**) stem cross-section, fragment; (**E**–**J**) leaves. Scales: a—1 mm for (**A**,**B**); b—50 µm for (**C**); c—200 µm for (**D**); d—1 mm for (**E**–**J**). All from V. Schiffner, 605 (FH).

**Figure 5 plants-13-00367-f005:**
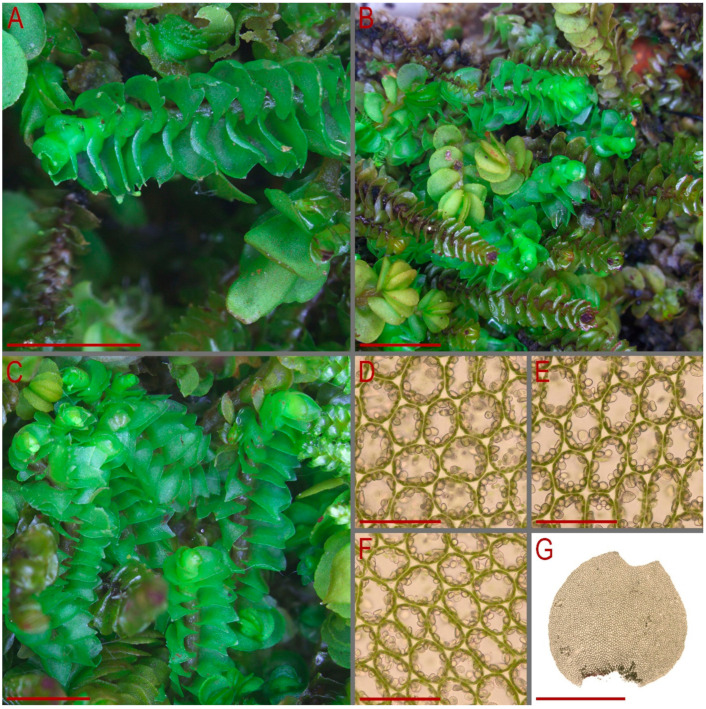
*Lophozia dubia* Shiffn. (**A**,**C**) part of mat.; (**B**) complex mat. (*Lophozia dubia* is intermixed here with *Anastrepta orcadensis* (Hook.) Schiffn. and *Mylia vietnamica* Bakalin & Vilnet); (**D**–**F**) midleaf cells with oil bodies; (**G**) leaf. Scales: 2 mm for (**A**–**C**); 50 µm for (**D**–**F**); 1 mm for (**G**). All from C-83-7-18 (VBGI).

**Figure 6 plants-13-00367-f006:**
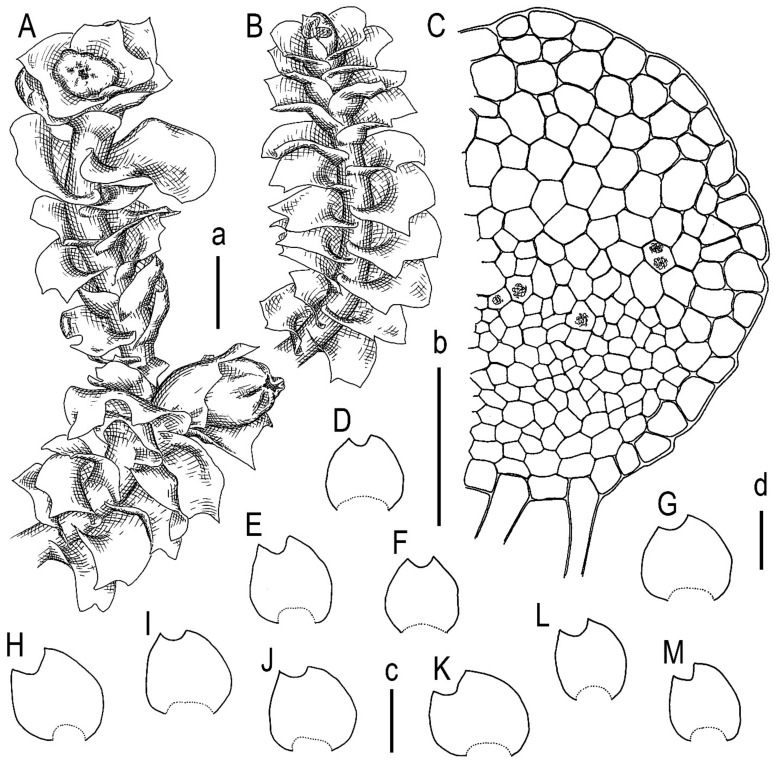
*Lophozia dubia* Schiffn. (**A**,**B**) upper part of a shoot fragment, dorsal view; (**C**) stem cross-section fragment; (**D**–**M**) leaves. Scales: a—1 mm for (**A**,**B**); b—100 µm for (**C**); c—1 mm for (**E**,**H**–**J**); d—1 mm for (**D**,**F**,**G**,**K**–**M**). All from C-83-7-18 (VBGI).

**Figure 7 plants-13-00367-f007:**
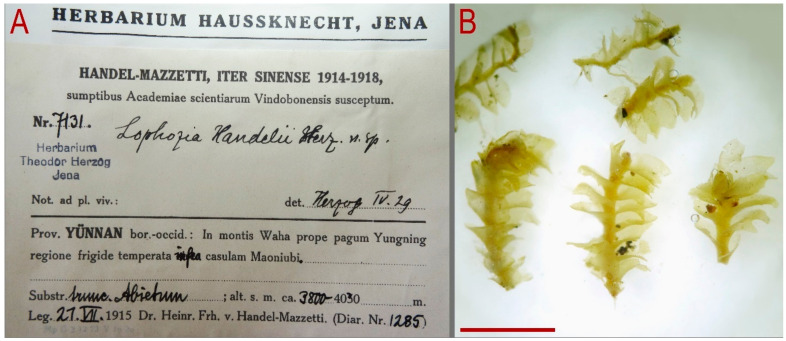
*Lophozia dubia* Schiffn. (**A**) label; (**B**) shoot fragments, dorsal view. Scale: 2 mm for (**B**). All from 7131 JE04009230 (lectotype of *Lophozia handelii* Herzog).

**Figure 8 plants-13-00367-f008:**
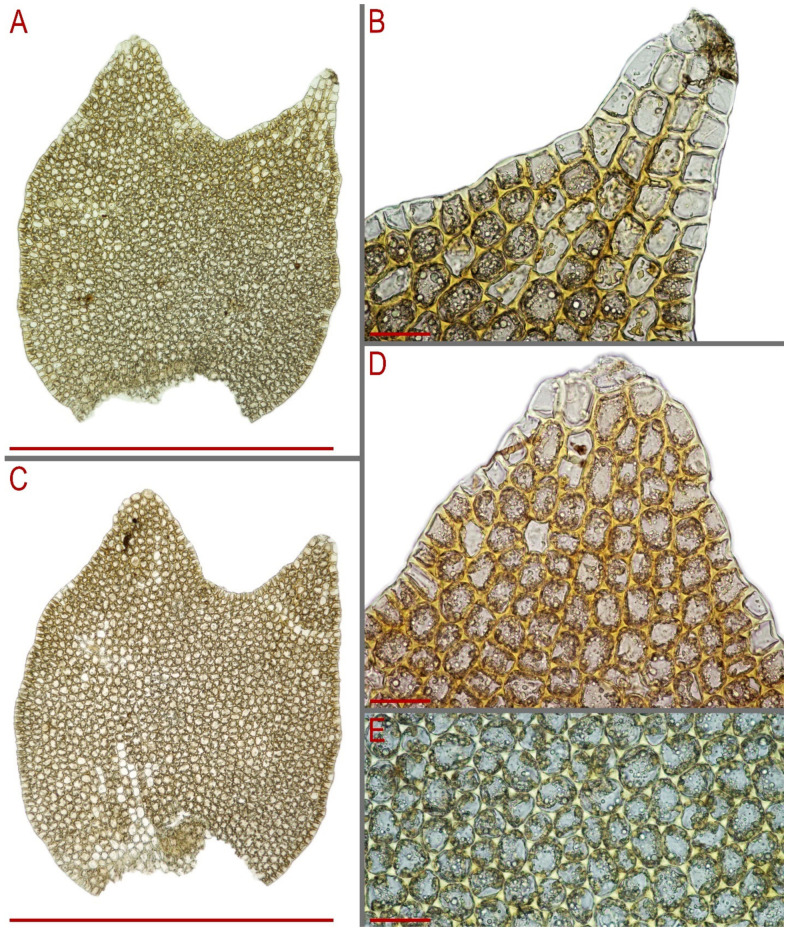
*Lophozia dubia* Schiffn. (**A**,**C**) leaves; (**B**,**D**) leaf apex cells; (**E**) midleaf cells with oil bodies. Scales: 1 mm for (**A**,**C**); 50 µm for (**B**,**D**,**E**). All from 7131 JE04009230 (lectotype of *Lophozia handelii* Herzog).

**Figure 9 plants-13-00367-f009:**
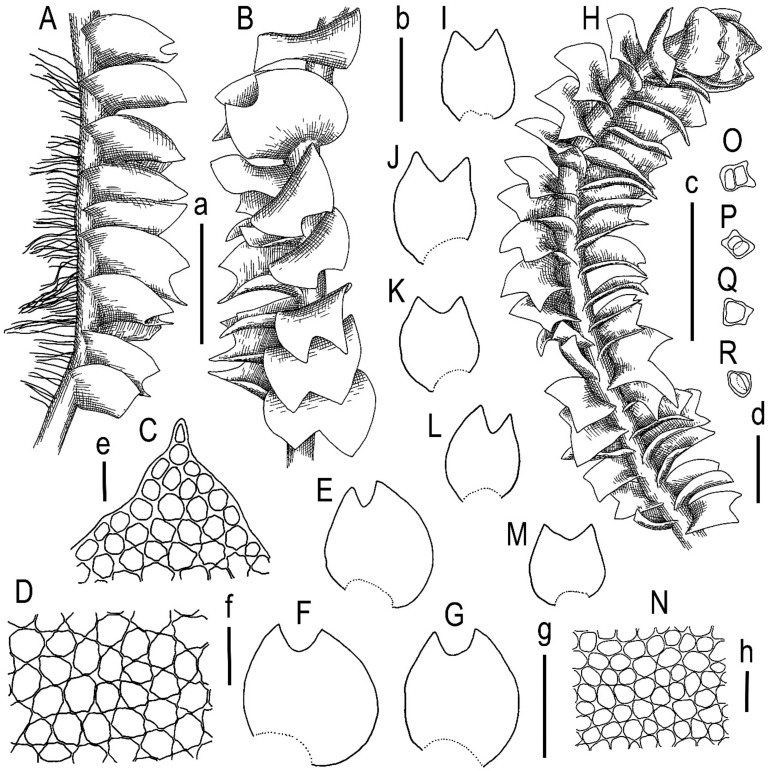
*Lophozia dubia* Schiffn. (**A**,**B**) middle part of a shoot fragment, side view; (**C**) leaf apex cells; (**D**) leaf middle cells; (**E**–**G**) leaves. All from G00053383/10810 (holotype of *Anastrophyllum pallidum* Steph.). (**H**) upper part of a shoot fragment, dorsal view; (**I**–**M**) leaves; (**N**) leaf middle cells; (**O**–**R**) gemmae. All from PC102533 (isolectotype of *Lophozia handelii* Herzog.). Scales: (a,c)—2 mm for (**A**,**B**,**H**); (b,g)—1 mm for (**E**–**G**,**I**–**M**); d,e,f,h—50 µm for (**C**,**D**,**N**,**O**–**R**).

**Figure 10 plants-13-00367-f010:**
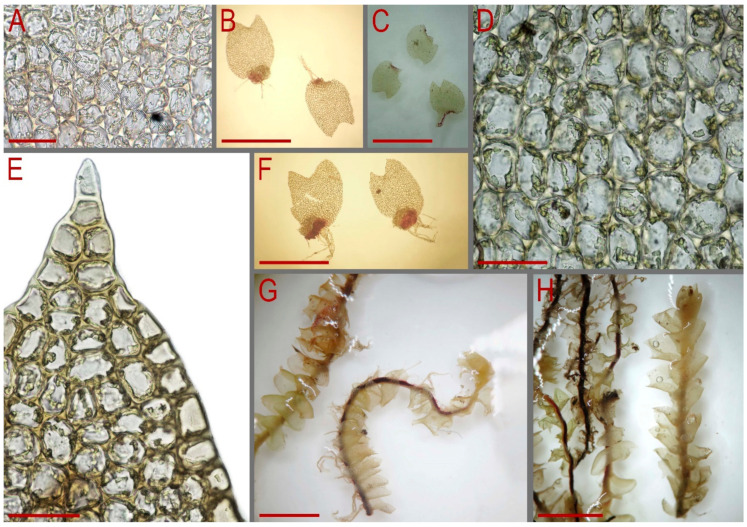
*Lophozia dubia* Schiffn. (**A**,**D**) leaf middle cells with oil bodies; (**B**,**C**,**F**) leaves; (**E**) leaf apex cells; (**G**,**H**) shoot fragments. Scales: 1 mm for (**B**,**F**); 2 mm for (**C**,**G**,**H**); 50 µm for (**A**,**D**,**E**). (**A**,**B**,**F**) from G00121692; (**C**–**E**,**G**,**H**) from Miehe 6993d.

**Figure 11 plants-13-00367-f011:**
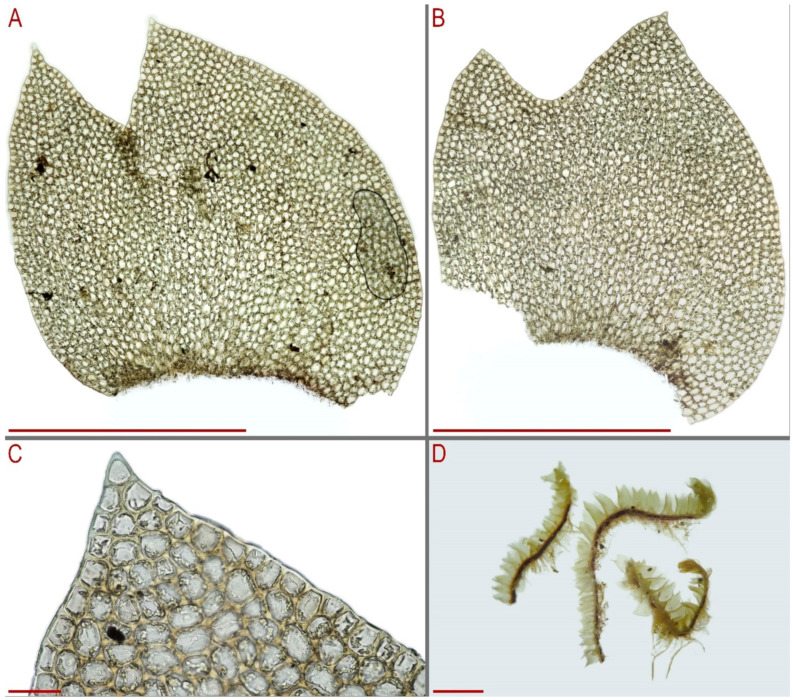
*Lophozia dubia* Schiffn. (**A**,**B**) leaves; (**C**) leaf apex cells; (**D**) shoot fragments. Scales: 500 µm for (**A**,**B**); 50 µm for (**C**); 2mm for (**D**). (**A**–**C**) from Miehe 6993d; (**D**) from Li X.J.920.

**Table 1 plants-13-00367-t001:** Infraspecific and interspecific *p*-distances over sequence pairs between groups. The number of base differences per site from averaging all sequence pairs between groups is shown. Calculation based on ITS1–2 (distances on lower left) and *trn*L–F (distances on upper right) nucleotide sequence data. n/c—not calculated.

№	Taxon	Infraspecific *p*-Distances, ITS1–2/*trn*L–F,	Interspecific *p*-Distances, ITS1–2/trnL–F,															
			1	2	3	4	5	6	7	8	9	10	11	12	13	14	15	16
1	*Lophozia dubia*	n/c/n/c		5.1	3.1	3.3	3.4	0.3	5.9	3.6	4.1	2.7	3.2	3.2	2.7	2.7	2.7	2.7
2	*Lophozia ascendens*	n/c/n/c	6.9		2.7	4.3	3.0	6.4	4.5	3.6	4.3	2.7	2.7	3.5	2.7	2.7	2.7	2.7
3	*Lophozia austrosibirica*	n/c/n/c	6.4	7.5		2.4	0.2	4.2	4.8	2.4	2.4	0.7	1.5	1.8	0.4	0.4	0.4	0.4
4	*Lophozia fuscovirens*	0.4/0	6.8	8.1	7.5		2.4	4.0	4.7	2.3	2.5	2.3	2.4	2.4	1.9	1.9	1.9	1.9
5	*Lophozia guttulata*	n/c/n/c	6.4	7.5	0	7.5		4.2	4.8	2.6	2.4	0.9	1.8	1.9	0.6	0.6	0.6	0.6
6	*Lophozia koreana*	0.5/0.2	5.2	8.0	8.4	7.6	8.4		6.4	4.4	4.9	3.8	3.8	3.8	3.7	3.7	3.7	3.7
7	*Lophozia lantratoviae*	0.8/0.2	5.9	5.8	6.1	6.8	6.1	6.2		5.9	4.9	4.7	4.4	5.7	4.6	4.8	4.6	4.6
8	*Lophozia longiflora*	n/c/n/c	4.5	5.4	5.6	4.1	5.6	5.2	4.1		3.7	2.7	3.0	1.8	2.4	2.4	2.4	2.4
9	*Lophozia pacifica*	n/c/n/c	6.4	7.3	2.2	7.4	2.2	7.8	5.8	5.2		2.3	1.6	3.3	2.0	2.0	2.0	2.0
10	*Lophozia schusteriana*	n/c/n/c	7.1	7.4	1.5	7.5	1.5	7.9	5.9	5.4	2.3		1.3	2.0	0.2	0.2	0.2	0.2
11	*Lophozia silvicola*	—/n/c	—	—	—	—	—	—	—	—	—	—		2.8	1.3	1.3	1.3	1.3
12	*Lophozia silvicoloides*	0/0	4.8	5.5	6.3	5.5	6.3	5.6	4.6	2.8	5.6	5.8	—		1.8	1.8	1.8	1.8
13	*Lophozia wenzelii*	n/c/n/c	7.6	8.2	1.7	8.3	1.7	8.8	6.4	6.1	2.5	1.5	—	6.6		0	0	0
14	*Lophozia wenzelii* var. *lapponica*	n/c/n/c	6.9	7.3	0.9	7.4	0.9	8.2	5.7	5.4	1.7	0.9	—	5.8	0.9		0	0
15	*Lophozia wenzelii* var. *litoralis*	n/c/n/c	6.9	7.8	0.9	7.8	0.9	8.2	6.1	5.5	2.2	1.7	—	6.2	1.9	1.0		0
16	*Lophozia wenzelii* var. *massularioides*	n/c/n/c	6.9	7.8	1.3	7.8	1.3	8.5	5.8	5.4	2.2	1.1	—	6.0	1.0	0.5	1.4	

**Table 2 plants-13-00367-t002:** The list of voucher details and GenBank accession numbers for the specimens used in the phylogenetic analysis in the present paper. The newly obtained sequences are marked in bold.

Specimen	Specimen Voucher	GenBank Accession Number	
		ITS1–2	*trn*L-*trn*F
*Lophozia dubia* Schiffn.	China, Yunnan Province, Diqing Prefecture, V.A. Bakalin & W.Z. Ma, C-83-7-18 (VBGI)	**OR982393**	**OR995729**
*Lophozia ascendens* (Warnst.) R.M. Schust.	Russia, Buryatiya, N.A. Konstantinova, 109-3-01 (KPABG)	DQ875089	DQ875054
*Lophozia austrosibirica* Bakalin	Russia, Buryatiya, V.A. Bakalin, 15-9-99 (KPABG)	DQ875105	DQ875069
*Lophozia fuscovirens* Bakalin & Vilnet	Russia, Magadan Prov., V.A. Bakalin, Mag-28-32-13 (KPABG)	MK007091	MK012211
*Lophozia fuscovirens* Bakalin & Vilnet	Russia, Magadan Prov., V.A. Bakalin, Mag-50-16-11 (KPABG)	MK007092	MK012212
*Lophozia fuscovirens* Bakalin & Vilnet	Russia, Magadan Prov., V.A. Bakalin, Mag-30-4-14 (KPABG)	MK007093	MK012213
*Lophozia fuscovirens* Bakalin & Vilnet	Norway, Svalbard, A. Savchenko, CA16-12-1c	MK774737	MK779914
*Lophozia guttulata* (Lindb. & Arnell) A. Evans	Russia, Buryatiya, N.A. Konstantinova, 81-1-01 (KPABG)	DQ875108	DQ875072
*Lophozia koreana* (Bakalin, S.S. Choi & B.Y. Sun) Maltseva, Vilnet & Bakalin	South Korea, Jeollabuk-do, V.A. Bakalin & S.S. Choi, Kor-74-5-19 (VGBI)	MW685441	MW654174
*Lophozia koreana* (Bakalin, S.S. Choi & B.Y. Sun) Maltseva, Vilnet & Bakalin	South Korea, Jeollabuk-do, V.A. Bakalin & S.S. Choi, Kor-75-15-19 (VGBI)	MW685442	MW654175
*Lophozia koreana* (Bakalin, S.S. Choi & B.Y. Sun) Maltseva, Vilnet & Bakalin	South Korea, Jeollabuk-do, V.A. Bakalin & S.S. Choi, Kor-76-1-19 (VGBI)	MW685443	MW654176
*Lophozia lantratoviae* Bakalin	Russia, Buryatiya, V.A. Bakalin, 76-7-01 (KPABG)	DQ875090	DQ875055
*Lophozia lantratoviae* Bakalin	China, Sichuan Prov., V.A. Bakalin, China-35-9-17, 37536 (VBGI)	MK007086	MK012206
*Lophozia lantratoviae* Bakalin	China, Sichuan Prov., V.A. Bakalin, China-39-3-17, 37283 (VBGI)	MK007087	MK012207
*Lophozia lantratoviae* Bakalin	Russia, Primorsky Terr., V.A. Bakalin, P-44-20-10 (KPABG)	MK007088	MK012208
*Lophozia lantratoviae* Bakalin	Russia, North Ossetia—Alanya Rep., A.V. Rumyantseva, 123153 (KPABG)	MW872337	MW887533
*Lophozia longiflora* (Nees) Schiffn.	Russia, Chitinskaya Prov, V.A. Bakalin, 11-5-00 (KPABG)	DQ875101	DQ875066
*Lophozia pacifica* Bakalin	Russia, Kamchatka Terr., V.A. Bakalin, K-16-2-02-VB, 103630 (KPABG)	MK007089	MK012209
*Lophozia schusteriana* Schljakov	Russia, Murmanskaya Prov, V.A. Bakalin, G9331 (KPABG)	DQ875103	DQ875067
*Lophozia silvicola* H. Buch	Russia, Sakhalin Prov., Sakhalin Isl., V.A. Bakalin, S-14-4-17 (VBGI)	MK007090	MK012210
*Lophozia silvicoloides* N. Kitag.	Russia, Kamchatskaya Prov, V.A. Bakalin, K-57-23-02-VB (KPABG)	DQ875098	DQ875063
*Lophozia silvicoloides* N. Kitag.	Russia, Murmanskaya Prov, N.A. Konstantinova, 356-4-00 (KPABG)	DQ875099	DQ875064
*Lophozia silvicoloides* N. Kitag.	Norway, Svalbard, N.A. Konstantinova, K241-1b-12 (KPABG)	MW298767	MW297148
*Lophozia wenzelii* (Nees) Steph.	Russia, Murmanskaya Prov, N.A. Konstantinova, 9329 (KPABG)	DQ875109	DQ875073
*Lophozia wenzelii* var. *lapponica* H. Buch & S.W. Arnell	Svalbard, Spitsbergen, N.A. Konstantinova, 124-2-04 (KPABG)	DQ875112	DQ875076
*Lophozia wenzelii* var. *litoralis* (Arnell) Bakalin	Russia, Murmanskaya Prov, V.A. Bakalin, 12-3-02 (KPABG)	DQ875110	DQ875074
*Lophozia wenzelii* var. *massularioides* Bakalin	Russia, Caucasus, Onipchenko, 31.08.83 (MHA)	DQ875111	DQ875075
*Lophoziopsis excisa* (Dicks.) Konstant. & Vilnet	Svalbard, Spitsbergen, N.A. Konstantinova, 104-1-04 (KPABG)	DQ875091	DQ875056
*Lophoziopsis excisa* (Dicks.) Konstant. & Vilnet	Russia, Murmanskaya Prov, N.A. Konstantinova, 41-2-97 (KPABG)	DQ875092	DQ875057
*Lophoziopsis excisa* (Dicks.) Konstant. & Vilnet	Svalbard, Spitsbergen, N.A. Konstantinova, K-21-2-05 (KPABG)	DQ875093	DQ875058
*Lophoziopsis excisa* (Dicks.) Konstant. & Vilnet	Russia, Maryi-El, N.A. Konstantinova, K437-2-04 (KPABG)	EF065691	EF065684
*Lophoziopsis excisa* (Dicks.) Konstant. & Vilnet	China, Sichuan Prov., V.A. Bakalin, China-37-2-17, 37462 (VBGI)	MK007094	MK012214
*Lophoziopsis excisa* (Dicks.) Konstant. & Vilnet	Norway, Svalbard, A. Savchenko, CA 364-2a-11 (KPABG)	MW298770	MW297151
*Lophoziopsis longidens* (Lindb.) Konstant. & Vilnet	Russia, Murmanskaya Prov, N.A. Konstantinova, 360-2-00 (KPABG)	DQ875094	DQ875059
*Lophoziopsis pellucida* (R.M. Schust.) Konstant. & Vilnet	Russia, Komi, M. Dulin, 103640 (KPABG)	EF065694	EF065686
*Lophoziopsis pellucida* (R.M. Schust.) Konstant. & Vilnet	Russia, Murmanskaya Prov., N.A. Konstantinova, 39-2a-03 (KPABG)	EF065695	EF065687
*Lophoziopsis pellucida* (R.M. Schust.) Konstant. & Vilnet	Russia, Magadan Prov., V.A. Bakalin, Mag-44-37-11, 115673 (KPABG)	MK007097	MK012217
*Lophoziopsis pellucida* (R.M. Schust.) Konstant. & Vilnet	Russia, Trans-Baikal Terr., Yu.S. Mamontov, 356-3-6 (KPABG)	MW298773	MW297154
*Lophoziopsis polaris* (R.M. Schust.) Konstant. & Vilnet	Russia, Magadan Prov., V.A. Bakalin, Mag-57-16-11, 115699 (KPABG)	MK007098	MK012218
*Lophoziopsis polaris* (R.M. Schust.) Konstant. & Vilnet	Norway, Svalbard, Konstantinova N.A., Savchenko A.N., K129-07 (KPABG)	MT334459	MT338482
*Lophoziopsis polaris* (R.M. Schust.) Konstant. & Vilnet	Norway, Svalbard, A.N. Savchenko, CA 19-29-3 (KPABG)	MW298774	MW297155
*Lophoziopsis polaris* var. *sphagnorum* (R.M. Schust.) Konstant. & Vilnet	Russia, Yakutiya, V.A. Bakalin, 23-11-00 (KPABG)	DQ875096	DQ875061
*Lophoziopsis propagulifera* (Gottsche) Konstant. & Vilnet	Russia, Kamchatskaya Prov, V.A. Bakalin, K-53-6-02-VB (KPABG)	DQ875097	DQ875062
*Obtusifolium obtusum* (Lindb.) S.W. Arnell	Sw-48-30-13 (VGBI)	MT504415	MT476337
*Trilophozia quinquedentata* (Huds.) Bakalin	Russia, Kareliya, Bakalin 2 July 1997	EU791804	AY327786
*Trilophozia quinquedentata* f. *gracilis* (R.M. Schust.) Konstant.	Svalbard, North-East Land, N.A. Konstantinova, K 118-2-06 (KPABG)	EU791802	EU791684
*Trilophozia quinquedentata* f. *gracilis* (R.M. Schust.) Konstant.	Svalbard, North-East Land, N.A. Konstantinova, K 72-2-06 (KPABG)	EU791803	EU791685
*Trilophozia quinquedentata* (Huds.) Bakalin	Norway, Svalbard, Konstantinova N.A., Savchenko A.N., K70-08 (KPABG)	MT334460	MT338484
*Tritomaria exsectiformis* (Breidl.) Schiffn. ex Loeske	Russia, Rep. Buryatiya, N.A. Konstantinova, 83-4-01 (KPABG)	EU791801	EU791683
*Tritomaria scitula* (Taylor) Jørg.	Russia, Komi Rep., M. Dulin, N.A. Konstantinova, G 101301 (KPABG)	EU791799	EU791681

**Table 3 plants-13-00367-t003:** Primers used in polymerase chain reaction (PCR) and cycle sequencing.

Locus	Sequence (5′-3′)	Direction	Annealing Temperature (°C)	Reference
*trn*L–F cpDNA	CGAAATTGGTAGACGCTGCG	forward	62	[20]
*trn*L–F cpDNA	TGCCAGAAACCAGATTTGAAC	reverse	58	[20]
ITS 1–2 nrDNA	ACCTGCGGAAGGATCATTG	forward	58	[37]
ITS 1–2 nrDNA	GATATGCTTAAACTCAGCGG	reverse	58	[38]

**Table 4 plants-13-00367-t004:** Protocols for PCRs.

Initial denaturation	3 min—94 °C	
Denaturation	30 s—95 °C	
Annealing	20 s (*trn*L–F), 30 s (ITS 1–2)	35 cycles
	at 58 °C (*trn*L–F), 60 °C (ITS 1–2)	
Elongation	30 s—72 °C	
Final elongation	3 min—72 °C	

## Data Availability

All data are contained within the article. New sequences have been deposited in the NCBI GenBank under accession numbers OR982393 and OR995729.

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
