# Peer review of "The Position of Lophozia dubia (Lophoziaceae, Marchantiophyta) in the Phylogenetic System of Lophozia and the Distribution of Lophozia in Southeast Eurasia, Extending to Indonesia"

_plants, 2024, doi:10.3390/plants13030367_

Round 1

Reviewer 1 Report

Comments and Suggestions for Authors

The manuscript brings a novel insights into understanding of complicated genus Lophozia. 

I suggest to avoid personal note in the criticism to other authors (Soderstrom, Vana, etc..) and their papers, but to keep neutral, justified and argumented scientific approach to the subject. Please change these sentences in Introduction chapter.

Some of the sentences from Introduction belongs to Results, and some from Discussion as well. e.g. how many species are recognized now. Please, check up. Avoid unnecessary repetitions.

In Concluding remarks genetic methods should be replaced by molecular methods.

I was not able to locate the sequence in GenBank, please provide the link.

Comments on the Quality of English Language

English is partly hard to follow, but generally OK. However, some improvement is needed.

Author Response

Thank you very much for constructive suggestions and comments. Below are our responses

Comment 1: I suggest to avoid personal note in the criticism to other authors (Soderstrom, Vana, etc..) and their papers, but to keep neutral, justified and argumented scientific approach to the subject. Please change these sentences in Introduction chapter.

Response: Agreed, all our opinion-based characteristics are deleted from the introduction section.

Comment 2: Some of the sentences from Introduction belongs to Results, and some from Discussion as well. e.g. how many species are recognized now. Please, check up. Avoid unnecessary repetitions.

Response: Agreed, there are several changes are made, we moved the couple of sentences from Introduction to results or conclusions and from Discussion to Results.

Comment 3: In Concluding remarks genetic methods should be replaced by molecular methods.

Response: Corrected in the Conclusion

Comment 4: I was not able to locate the sequence in GenBank, please provide the link.

Response: The scheduled release date for our submission to GenBank is Jan 1, 2025. In accordance with the GenBank guideline the sequences will be released when the article citing this accession numbers are published or on the above release date, whichever comes first.

Comment 4: English is partly hard to follow, but generally OK. However, some improvement is needed.

Response: Some improvements were made by English speaking peer.

Reviewer 2 Report

Comments and Suggestions for Authors

The paper is very well organized and the molecular studies are particularly in-depth and effectively show the correlations between the different taxa studied. He hopes to expand studies on other taxa belonging to the same genus.

Author Response

Comment The paper is very well organized and the molecular studies are particularly in-depth and effectively show the correlations between the different taxa studied. He hopes to expand studies on other taxa belonging to the same genus.

Response: We are very grateful for your attention to our manuscript and its high evaluation